# HAS MY SYSTEM PROMPT BEEN USED? LARGE LANGUAGE MODEL PROMPT MEMBERSHIP INFERENCE

## ABSTRACT

Prompt engineering has emerged as a powerful technique for optimizing large language models (LLMs) for specific applications, enabling faster prototyping and improved performance, and giving rise to the interest of the community in protecting proprietary system prompts. In this work, we explore a novel perspective on prompt privacy through the lens of membership inference. We develop Prompt Detective, a statistical method to reliably determine whether a given system prompt was used by a third-party language model. Our approach relies on a statistical test comparing the distributions of two groups of model outputs corresponding to different system prompts. Through extensive experiments with a variety of language models, we demonstrate the effectiveness of Prompt Detective for prompt membership inference. Our work reveals that even minor changes in system prompts manifest in distinct response distributions, enabling us to verify prompt usage with statistical significance.

## 1 INTRODUCTION

Prompt engineering offers a powerful, flexible, and fast way to optimize large language models (LLMs) for specific applications, enabling faster and cheaper customization than finetuning while delivering strong specialized performance. Large language model providers, such as Anthropic and OpenAI, release detailed prompt engineering guides on prompting strategies allowing their customers to reduce hallucination rates and optimize business performance (OpenAI, 2023; Anthropic, 2024b). The use of system prompts also provides specialized capabilities such as taking on a character

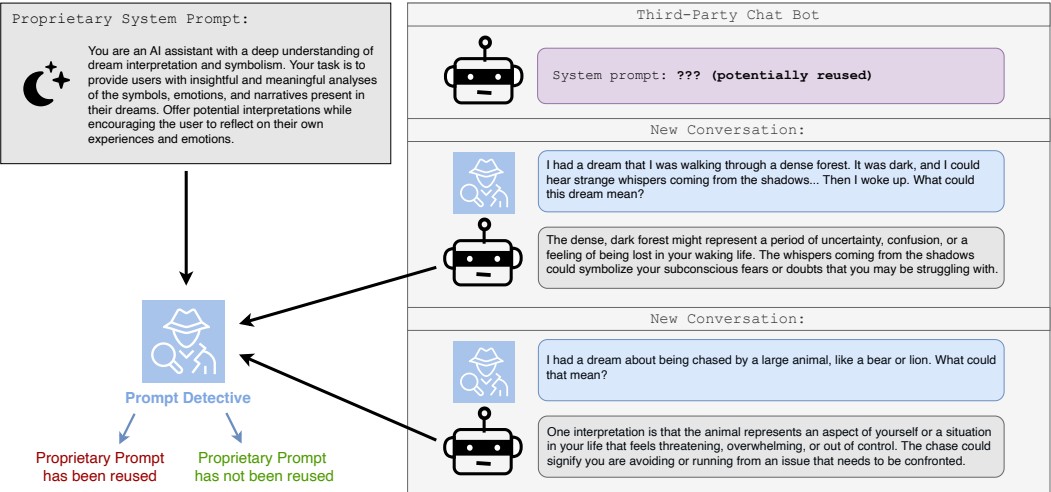

Figure 1: **Prompt Detective** verifies if a third-party chat bot uses a given proprietary system prompt by querying the system and comparing distribution of outputs with outputs obtained using proprietary system prompt.

which is often leveraged by startups in their products[1]. Developers put significant effort into prompt engineering and prompts optimized for specific use-cases are even sold at online marketplaces[2].

The importance and promise of prompt engineering gave rise to the interest of the community in protecting proprietary prompts and a growing body of academic literature explores prompt reconstruction attacks (Hui et al., 2024; Zhang et al.; Morris et al., 2023; Geiping et al., 2024) which attempt to recover a prompt used in a language model to produce particular generations. These methods achieve impressive results in approximate prompt reconstruction, however their reconstruction success rate is not high enough to be able to confidently verify the prompt reuse, they are computationally expensive usually relying on GCG-style optimization (Zou et al., 2023), and some of these methods require access to model gradients (Geiping et al., 2024). Additionally, while some reconstruction methods provide confidence scores (Zhang et al.), they do not offer statistical guarantees for prompt usage verification.

In this work, we specifically focus on the problem of verifying if a particular system prompt was used in a large language model. This problem can be viewed through the lens of an adversarial setup: an attacker may have reused someone else's proprietary system prompt and deployed an LLM-based chat bot with it. In LLM-based chatbots, control over part of the input is given to the end user. Consider a customer service chatbot that employs a general LLM to help customers get the answers they need. These systems typically add the user input into a longer template that includes a system prompt with application-specific instructions to help ensure that the back end large general purpose language model returns useful content to the user. Note the value in an expertly written system prompt – it could be critical in getting quality responses from large back end LLMs. Assuming access to querying this chatbot, can we verify with statistical significance if the proprietary system prompt has not been used? In other words, we develop a method for system prompt membership inference. Our contributions are as follows:

- We develop Prompt Detective, a training-free statistical method to reliably verify whether a given system prompt was used by a third-party language model, assuming query access to it.
- We extensively evaluate the effectiveness of Prompt Detective across a variety of language models, including Llama, Mistral, Claude, and GPT families including challenging scenarios such as distinguishing similar system prompts and black-box settings.
- Our work reveals that even minor changes in system prompts manifest in distinct response distributions of LLMs, enabling Prompt Detective to verify prompt usage with statistical significance. This highlights that LLMs take specific trajectories when generating responses based on the provided system prompt.

## 2 RELATED WORK

### 2.1 PROMPT EXTRACTION ATTACKS

Prompt engineering has emerged as an accessible approach to adapt LLMs for specific user needs (Liu et al., 2023), with system prompts playing a crucial role in shaping LLM outputs and driving performance across application domains (Ng & Fulford, 2023). Prior work has proposed several prompt extraction attacks, which deduce the content of a proprietary system prompt by interacting with a model, both for language models (Morris et al., 2023; Zhang et al.; Sha & Zhang, 2024; Yang et al., 2024) and for image generation models (Wen et al., 2024). Morris et al. (2023) frame the problem as model inversion, where they deduce the prompt given next token probabilities. Similarly, Sha & Zhang (2024) propose a method to extract prompts from sampled generative model outputs. Furthermore, Yang et al. (2024) describe a way to uncover system prompts using context and response pairs. Additionally, Zhang et al. present an evaluation of prompt extraction attacks for a variety of modern LLMs. In contrast to the works on inversion style methods, one can also find adversarial inputs that jailbreak LLMs (Zou et al., 2023; Cherepanova & Zou, 2024; Geiping et al., 2024) and even lead them to eliciting the system prompt in the response. Prompt reconstruction methods can also be adapted to solve the problem of prompt verification through comparing the reconstructed prompt to the reference prompt, however, their high computational cost (Hui et al., 2024; Geiping et al., 2024), the need to access model gradients (Geiping et al., 2024), and

---

[1]https://character.ai/

[2]https://prompti.ai/chatgpt-prompt, https://promptbase.com/.

imperfect reconstruction success rate (Hui et al., 2024; Zhang et al.; Geiping et al., 2024) motivate the development of methods specifically tailored to the problem of prompt reuse verification.

## 2.2 DATA MEMBERSHIP INFERENCE AND EXTRACTION ATTACKS ON LANGUAGE MODELS

In the evolving discussion on data privacy, a significant topic is membership inference, which involves determining whether a particular data point is part of a model's training set (e.g. Yeom et al., 2018; Sablayrolles et al., 2019; Salem et al., 2018; Song & Mittal, 2021; Hu et al., 2022). Shokri et al. (2017) and Carlini et al. (2022) both propose methods to determine membership in the training data based on the idea that models tend to behave differently on their training data than on other data. Bertran et al. (2024) further propose a more effective method and alleviate the need to know the target model's architecture, while Wen et al. (2022) propose perturbing the query data to improve accuracy of their attack. Compared to the standard membership inference setting, our work addresses a related but distinct question: whether a given text is part of the LLM input context, thus exploring prompt membership inference.

## 3 PROMPT DETECTIVE

### 3.1 SETUP

Prompt Detective aims to verify whether a particular known system prompt is used by a third-party chat bot as shown in Figure 1. In our setup, we assume an API or online chat access to the model, that is, we can query the chat bot with different task prompts and we have control over choosing these task prompts. We also assume the knowledge about which model is employed by the service in most of our experiments, and we explore the black-box scenario in section 6.

This setup can be applied when a user, who may have spent significant effort developing the system prompt for their product such as an LLM character or a domain-specific application, suspects that their proprietary system prompt has been utilized by a third-party chat service effectively replicating the behavior of their product, and wants to verify if that was in fact the case while only having online chat window access to that service.

Moreover, this adversarial setup can be seen through the lens of membership inference attacks, where instead of verifying membership of a given data sample in the training data of a language model, we verify membership of a particular system prompt in the context window of a language model. We therefore refer to our adversarial setting as *prompt membership inference*.

### 3.2 HOW DOES IT WORK?

Let $f$ denote a language model, let $p$ be a system prompt, and let $q$ be a task prompt. Together, we denote the full output as $f_p(q)$. For example, a system prompt could look like "You are a helpful assistant" and a task-specific query might be like "Can you help me with a billing issue?" If we applied the appropriate chat template, the full string we pass to the model's tokenizer would look as follows:

```
[SYS] You are a helpful assistant [\SYS]

[USER] Can you help me with a billing issue?[\USER]
```

We assume the model owner uses an unknown system prompt $p$, and that we can query the service with task prompts $q$ to get output $f_p(q)$. We also assume access to a similar model prompted with our known proprietary system prompt $\bar{p}$, whose output is denoted by $\bar{f}\bar{p}$. Our goal is to determine whether $p$ and $\bar{p}$ are distinct.

**Core idea.** Prompt Detective is a training-free statistical method designed specifically for determining if a system prompt used in an LLM-based service matches a known string. The core idea is to compare the distributions of two groups of generations corresponding to different system prompts and apply a statistical test to assess if the distributions are significantly different, which would indicate that the system prompts are distinct. That is, Prompt Detective compares the distributions of

---

**Algorithm 1** Prompt Detective

---

**Require:** Third-party language model $f_p$,
         Known (proprietary) system prompt $\bar{p}$,
         Model $f_{\bar{p}}$,
         Task prompts $q_1, \ldots, q_n$,
         Number of responses per task prompt $k$,
         Significance level $\alpha$

$G_1 \leftarrow \{\{f_p(q_1)^1...f_p(q_1)^k\}, \ldots, \{f_p(q_n)^1...f_p(q_n)^k\}\}$   ▷ *Generations from third-party model*
$G_2 \leftarrow \{\{f_{\bar{p}}(q_1)^1...f_{\bar{p}}(q_1)^k\}, \ldots, \{f_{\bar{p}}(q_n)^1...f_{\bar{p}}(q_n)^k\}\}$   ▷ *Generations from known prompt*
$V_1 \leftarrow \text{BERT}(G_1)$   ▷ *BERT embeddings of $G_1$*
$V_2 \leftarrow \text{BERT}(G_2)$   ▷ *BERT embeddings of $G_2$*
$\mu_1 \leftarrow \text{Mean}(V_1), \mu_2 \leftarrow \text{Mean}(V_2)$   ▷ *Mean vectors*
$s_{\text{obs}} \leftarrow \text{CosineSimilarity}(\mu_1, \mu_2)$   ▷ *Observed cosine similarity*
$c \leftarrow 0$   ▷ *Counter for extreme cosine similarities*
**for** $i = 1$ to $N_{\text{permutations}}$ **do**   ▷ *Permutation test loop*
     $V_1^* \leftarrow V_1, V_2^* \leftarrow V_2$   ▷ *Initialize permuted groups*
     **for** $j = 1$ to $n$ **do**   ▷ *Shuffle preserving the task prompt structure*
         $V_{\text{combined}} \leftarrow V_1^*[(j-1)k : jk] \cup V_2^*[(j-1)k : jk]$   ▷ *Concatenate responses*
         $V_{\text{combined}} \leftarrow \text{Shuffle}(V_{\text{combined}})$   ▷ *Permute combined responses*
         $V_1^*[(j-1)k : jk] \leftarrow V_{\text{combined}}[: k]$   ▷ *Assign first part to $V_1^*$*
         $V_2^*[(j-1)k : jk] \leftarrow V_{\text{combined}}[k :]$   ▷ *Assign second part to $V_2^*$*
     $\mu_1^* \leftarrow \text{Mean}(V_1^*), \mu_2^* \leftarrow \text{Mean}(V_2^*)$
     $s^* \leftarrow \text{CosineSimilarity}(\mu_1^*, \mu_2^*)$
     **if** $s^* \leq s_{\text{obs}}$ **then**   ▷ *Check if new similarity is as extreme*
         $c \leftarrow c + 1$   ▷ *Increment counter for extreme similarities*
$p \leftarrow c/N_{\text{permutations}}$
**if** $p < \alpha$ **then**
     **return** "Prompts are distinct"
**else**
     **return** "Insufficient evidence to claim prompts are distinct"

---

high-dimensional vector representations of two groups of generations

$$\{f_p(q_i)^j\}_{i \in [1,\ldots,n], j \in [1,\ldots,k]} \quad \text{and} \quad \{\bar{f}_{\bar{p}}(q_i)^j\}_{i \in [1,\ldots,n], j \in [1,\ldots,k]},$$

where the first set of generations is obtained from the third-party service $f_p$ prompted with task queries $q_i$ (with $k$ responses sampled for each task query), and the second set of generations is obtained from the $\bar{f}\bar{p}$ model prompted with the proprietary prompt $\bar{p}$ and the same task queries.

**Text representations.** We use BERT embedding (Reimers & Gurevych, 2019) to map strings to representation vectors. We compute the BERT embeddings for both $\{f_p(q_i)^j\}_{i \in [1,\ldots,n], j \in [1,\ldots,k]}$ and $\{\bar{f}_{\bar{p}}(q_i)^j\}_{i \in [1,\ldots,n], j \in [1,\ldots,k]}$, yielding two groups of high-dimensional vector representations of generations corresponding to the two system prompts under comparison. We include results for ablation study on embedding models in Appendix B Table 4.

**Statistical test of the equality of representation distributions.** To compare the distributions of these two groups, we employ a permutation test (Good, 2013) with the cosine similarity between the mean vectors of the groups used as the test statistic. The permutation test is a non-parametric approach that does not make assumptions about the underlying distribution of the data, making it a suitable choice for Prompt Detective. Intuitively, the permutation test assesses whether the observed difference between the two groups of generations is significantly larger than what would be expected by chance if the generations were not influenced by the underlying system prompts. By randomly permuting the responses within each task prompt across the two groups, the test generates a null distribution of cosine similarities between their mean vectors under the assumption that the system prompts are identical, while preserving the task prompt structure. The observed cosine similarity is then compared against this null distribution to determine its statistical significance. Algorithm 1 outlines all of the steps of Prompt Detective in detail.

**Original System Prompt:**
Imagine you are an alien anthropologist studying human culture and customs. Analyze the following aspects of human society from an objective, outsider's perspective. Provide detailed observations, insights, and hypotheses based on the available information.

**Same Prompt, Minimal Rephrasing (Similarity Level 1):**
Adopt the perspective of an alien anthropologist studying human culture and customs. Analyze the provided aspects of human society from an objective, outsider's viewpoint. Offer detailed observations, insights, and hypotheses based on the available information.

**Same Prompt, Minor Rephrasing (Similarity Level 2):**
Take on the role of an alien anthropologist tasked with studying human culture and customs. From an objective, outsider's perspective, analyze the given aspects of human society. Provide detailed observations, insights, and hypotheses based on the information available to you.

**Same Prompt, Significant Rephrasing (Similarity Level 3):**
Imagine yourself as an alien anthropologist conducting a study on human culture and customs. From an objective, outsider's perspective, analyze the specified aspects of human society. Offer detailed observations, insights, and hypotheses based on the information provided to you.

**Different Prompt, Remote Similarities (Similarity Level 4):**
As an intergalactic cultural researcher, your task is to observe and analyze various aspects of human society from an objective, outsider's perspective, as if you were an alien anthropologist studying an unfamiliar civilization. Provide detailed observations, insights, and hypotheses based on the information available to you.

**Different Prompt, Significant Conceptual Changes (Similarity Level 5):**
You are an AI anthropologist from an advanced alien civilization, tasked with conducting an in-depth study of human culture and customs. Approach the provided aspects of human society from an objective, outsider's perspective, free from preconceptions or biases. Offer detailed observations, insights, and hypotheses based on the available information, as if you were encountering this civilization for the first time.

Figure 2: **Hard Examples** illustrate varying degrees of similarity between the original prompts and their rephrased versions. Similarity Level 1 is highly similar, while Level 5 is completely different.

### 3.3 TASK QUERIES

The selection of task prompts $q_1, \ldots, q_n$ is an important component of Prompt Detective, as these prompts serve as probes to elicit responses that are influenced by the underlying system prompt. Since we assume control over the task prompts provided to the third-party chat bot, we can strategically choose them to reveal differences in the response distributions induced by distinct system prompts.

We consider a task prompt a good probe for a given system prompt if it elicits responses that are directly influenced by and related to the system prompt. For example, if the system prompt is designed for a particular LLM persona or role, task prompts that encourage the model to express its personality, opinions, or decision-making processes would be effective probes. A diverse set of task prompts can be employed to increase the robustness of Prompt Detective. In practice, we generated task queries for each of the system prompts $\bar{p}$ in our experiments with the Claude 3 Sonnet (Anthropic, 2024a) language model unless otherwise noted (see Appendix F).

## 4 EXPERIMENTAL SETUP

### 4.1 SYSTEM PROMPT SOURCES

**Awesome-ChatGPT-Prompts** [3] is a curated collection of 153 system prompts that enable users to tailor LLMs for specific roles. This dataset includes prompts for creative writing, programming, productivity, etc. Prompts are designed for various functions, such as acting as a Startup Idea Generator, Python Interpreter, or Personal Chef. The accompanying task prompts were generated with Claude 3 Sonnet (see Appendix **??**). For the 153 system prompts in Awesome-ChatGPT, we generated overall 50 task prompts. In these experiments, while a given task prompt is not necessarily a good probe for every system prompt, these 50 task prompts include at least one good probe for each of the system prompts.

**Anthropic's Prompt Library** [4] provides detailed prompts that guide models into specific characters and use cases. For our experiments, we select all of the personal prompts from the library that include system prompts giving us 20 examples. Personal prompts include roles such as Dream Interpreter or Emoji Encoder. As the accompanying task prompts, we used 20 of the corresponding user prompts provided in the library.

---

[3]https://github.com/f/awesome-chatgpt-prompts
[4]https://docs.anthropic.com/en/prompt-library/library

**Hard Examples:** To evaluate the robustness of Prompt Detective in challenging scenarios, we create a set of hard examples by generating variations of prompts from Anthropic's Prompt Library. These variations are designed to have different levels of similarity to the original prompts, ranging from minimal rephrasing to significant conceptual changes, producing varying levels of difficulty for distinguishing them from the original prompts.

For each system prompt from Anthropic's Prompt Library, we generate five variations with the following similarity levels (see Figure 2 for examples):

1. **Same Prompt, Minimal Rephrasing**: The same prompt, slightly rephrased with minor changes in a few words.
2. **Same Prompt, Minor Rephrasing**: Very similar in spirit, but somewhat rephrased.
3. **Same Prompt, Significant Rephrasing**: Very similar in spirit, but significantly rephrased.
4. **Different Prompt, Remote Similarities**: A different prompt for the same role with some remote similarities to the original prompt.
5. **Different Prompt, Significant Conceptual Changes**: A completely different prompt for the same role with significant conceptual changes.

This process results in a total of 120 system prompts for hard examples. The system prompt variations and the accompanying task prompts were generated with the Claude 3 Sonnet model. For the hard example experiments, we generated 10 specific probe task queries per each of the original system prompts (see Appendices A,**??**).

## 4.2 MODELS

We conduct our experiments with a variety of open-source and API-based models, including Llama2 13B (Touvron et al., 2023), Llama3 70B [5], Mistral 7B (Jiang et al., 2023), Mixtral 8x7B (Jiang et al., 2024), Claude 3 Haiku (Anthropic, 2024a), and GPT-3.5 (Achiam et al., 2023).

## 4.3 EVALUATION: STANDARD AND HARD EXAMPLES

In the standard setup, to evaluate Prompt Detective, we construct pairs of system prompts representing two scenarios: (1) where the known system prompt $\bar{p}$ is indeed used by the language model (positive case), and (2) where the known system prompt $\bar{p}$ differs from the system prompt $p$ used by the model (negative case). The positive case simulates a situation where the proprietary prompt has been reused, while the negative case represents no prompt reuse.

We construct a positive pair $(\bar{p}, \bar{p})$ for each of the system prompts and randomly sample the same number of negative pairs $(\bar{p}, p), \bar{p} \neq p$. The negative pairs may not represent similar system prompts, and we refer to this setting as the standard setup.

For the hard example setup, we construct prompt pairs using the variations of the Anthropic Prompt Library prompts with different levels of similarity, as described in section 4.1. The first prompt in each pair is the original prompt from the library, while the second prompt is one of the five variations, ranging from minimal rephrasing to significant conceptual changes. That is, while in this setup there are no positive pairs using identical prompts, some of the pairs represent extremely similar prompts differing by only very few words replaced with synonyms.

## 5 RESULTS

### 5.1 PROMPT DETECTIVE CAN DISTINGUISH SYSTEM PROMPTS

Table 1 shows the effectiveness of Prompt Detective in distinguishing between system prompts in the standard setup across different models and prompt sources. We report the false positive rate (FPR) and false negative rate (FNR) at a standard $p$-value threshold of 0.05, along with the average $p$-value for both positive and negative prompt pairs. In all models except for Claude on

---

[5]https://ai.meta.com/blog/meta-llama-3/

Table 1: **Prompt Detective** can reliably detect when system prompt used to produce generations is different from the given proprietary system prompt. We report false positive and false negative rates at a standard 0.05 $p$-value threshold. Additionaly, we report average $p$-value for positive and negative system prompt pairs.

| | Awesome-ChatGPT-Prompts | | | | Anthropic Library | | | |
|---|---|---|---|---|---|---|---|---|
| | FPR | FNR | $p_{avg}^p$ | $p_{avg}^n$ | FPR | FNR | $p_{avg}^p$ | $p_{avg}^n$ |
| Llama2 13B | 0.00 | 0.05 | $0.491_{\pm.28}$ | $0.000_{\pm.00}$ | 0.00 | 0.10 | $0.483_{\pm.30}$ | $0.000_{\pm.00}$ |
| Llama3 70B | 0.00 | 0.07 | $0.484_{\pm.29}$ | $0.000_{\pm.00}$ | 0.00 | 0.00 | $0.508_{\pm.29}$ | $0.000_{\pm.00}$ |
| Mistral 7B | 0.00 | 0.04 | $0.503_{\pm.29}$ | $0.000_{\pm.00}$ | 0.00 | 0.05 | $0.581_{\pm.33}$ | $0.000_{\pm.00}$ |
| Mixtral 8x7B | 0.00 | 0.03 | $0.475_{\pm.30}$ | $0.000_{\pm.00}$ | 0.00 | 0.00 | $0.466_{\pm.30}$ | $0.000_{\pm.00}$ |
| Claude Haiku | 0.05 | 0.03 | $0.543_{\pm.29}$ | $0.021_{\pm.11}$ | 0.00 | 0.05 | $0.440_{\pm.28}$ | $0.000_{\pm.00}$ |
| GPT-3.5 | 0.00 | 0.06 | $0.501_{\pm.28}$ | $0.000_{\pm.00}$ | 0.00 | 0.00 | $0.396_{\pm.26}$ | $0.000_{\pm.00}$ |

AwesomeChatGPT dataset, Prompt Detective consistently achieves a zero false positive rate, and the false negative rate remains approximately 0.05. This rate corresponds to the selected significance level, indicating the probability of Type I error – rejecting the null hypothesis that system prompts are identical when they are indeed the same. Figure 3 shows how the average $p$-value changes in negative cases (where the prompts differ) as the number of task queries increases. As expected, the $p$-value decreases with more queries, providing stronger evidence for rejecting the null hypothesis of equal distributions. Consequently, increasing the number of queries further improves the statistical test's power, allowing for the use of lower significance levels and thus ensuring a reduced false negative rate, while maintaining a low false positive rate.

While there are no existing prompt membership inference baselines, prompt reconstruction methods can be adapted to the prompt membership inference setting by comparing recovered system prompts to the reference system prompts. We compare PLeak (Hui et al., 2024) – one of the most high performing of the existing prompt reconstruction approaches to Prompt Detective in the prompt membership setting. We find Prompt Detective to be significantly more effective in the prompt membership inference setting and report the results in Table 5 of Appendix B.1.

## 5.2 Hard examples: similar system prompts

Table 2 presents the results for the challenging hard example setup, where we evaluate Prompt Detective's performance on system prompts with varying degrees of similarity to the proprietary prompt. We conduct this experiment with Claude 3 Haiku and GPT-3.5 models, testing Prompt Detective in two scenarios. First, we use 2 generations per task prompt, resulting in 20 generations for each system prompt, as in the standard setup Anthropic Library experiments. Second, we use 50

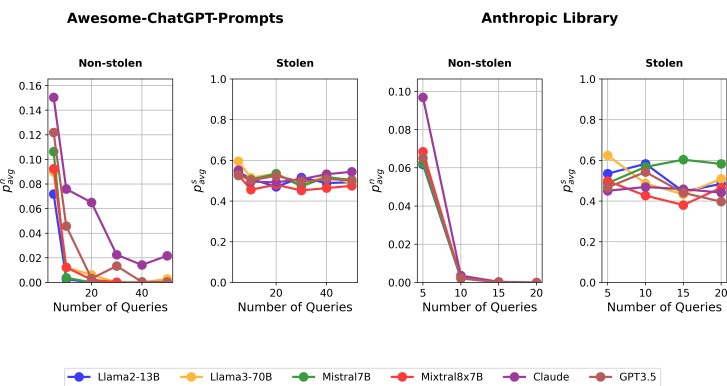

Figure 3: **Average $p$-value computed for different number of task queries. Left: Awesome-ChatGPT-Prompts. Right: Anthropic Library.** Increasing the number of generations leads to decreasing $p$-value in negative cases, but the average $p$-value for positive cases remains close to 0.5.

Table 2: **Results for Hard Examples.** Increasing similarity between the proprietary system prompt and prompt used in third-party system (lower similarity level) leads to worse separation of generation distributions. Subscript in model name corresponds to the number of generations per task prompt used in Prompt Detective.

| Model | Similarity 1 | | Similarity 2 | | Similarity 3 | | Similarity 4 | | Similarity 5 | |
|---|---|---|---|---|---|---|---|---|---|---|
| | $p_{avg}$ | FPR | $p_{avg}$ | FPR | $p_{avg}$ | FPR | $p_{avg}$ | FPR | $p_{avg}$ | FPR |
| Claude$_2$ | $0.194_{\pm.22}$ | 0.65 | $0.108_{\pm.19}$ | 0.35 | $0.093_{\pm.25}$ | 0.15 | $0.052_{\pm.18}$ | 0.10 | $0.052_{\pm.13}$ | 0.20 |
| Claude$_{50}$ | $0.007_{\pm.03}$ | 0.05 | $0.000_{\pm.00}$ | 0.00 | $0.000_{\pm.00}$ | 0.00 | $0.000_{\pm.00}$ | 0.00 | $0.000_{\pm.00}$ | 0.00 |
| GPT-3.5$_2$ | $0.213_{\pm.25}$ | 0.65 | $0.306_{\pm.34}$ | 0.60 | $0.225_{\pm.26}$ | 0.60 | $0.050_{\pm.10}$ | 0.20 | $0.020_{\pm.04}$ | 0.10 |
| GPT-3.5$_{50}$ | $0.000_{\pm.00}$ | 0.00 | $0.011_{\pm.05}$ | 0.05 | $0.000_{\pm.00}$ | 0.00 | $0.000_{\pm.00}$ | 0.00 | $0.000_{\pm.00}$ | 0.00 |

generations for each task query, resulting in 500 generations per system prompt in total. We observe that when only 2 generations are used, the false positive rate is high reaching 65% for GPT 3.5 and Claude models in Similarity Level 1 setup, indicating the challenge of distinguishing the response distributions for two very similar system prompts. However, increasing the number of generations for each probe to 50 leads to Prompt Detective being able to almost perfectly separate between system prompts even in the highest similarity category.

We further explore the effect of including more generations and more task prompts on Prompt Detective's performance. In Figure 4, we display the average $p$-value for Prompt Detective on Similarity Level 1 pairs versus the number of generations, the number of task prompts, and the number of tokens in the generations. We ask the following question: for a fixed budget in terms of the total number of tokens generated, is it more beneficial to include more different task prompts, more generations per task prompt, or longer responses from the model? Our observations suggest that while having more task prompts is comparable to having more generations per task prompt, it is important to have at least a few different task prompts for improved robustness of the method. However, having particularly long generations exceeding 64 tokens is not as useful, indicating that the optimal setup includes generating shorter responses to more task prompts and including more generations per task prompt.

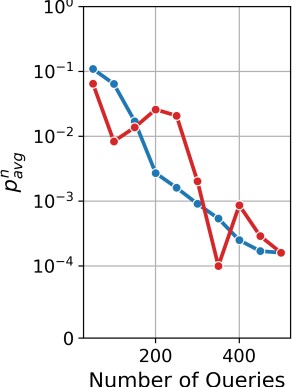 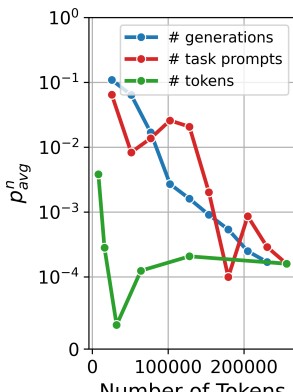

Figure 4: **Effect of the number of task prompts, generations, and tokens on the performance of Prompt Detective.** Average $p$-value for GPT-3.5 model on system prompts of Similarity Level 1. The left panel shows the average $p$-value vs. the number of generations used in Prompt Detective. The blue line represents results with 10 task prompts and 5–50 generations (512 tokens long) per prompt. The red line represents results with 1–10 task prompts, each with 50 generations (512 tokens long). The right panel plots $p_{avg}^n$ against the total number of tokens generated, with the green line showing results using 10 task prompts and 50 shorter generations (16–512 tokens long)

Table 3: **Prompt Detective in Black Box Setup.** Assuming the third-party model $f_p$ is one of the six models from previous experiments, we use Prompt Detective to compare it against each of the six reference models $\{\bar{f}_p^i\}_{i=1}^6$.

| Model | Awesome-ChatGPT-Prompts | | | | Anthropic Library | | | |
|---|---|---|---|---|---|---|---|---|
| | FPR | FNR | $p_{avg}^p$ | $p_{avg}^n$ | FPR | FNR | $p_{avg}^p$ | $p_{avg}^n$ |
| Llama2 13B | 0.00 | 0.01 | $0.493_{\pm.28}$ | $0.000_{\pm.00}$ | 0.00 | 0.05 | $0.484_{\pm.30}$ | $0.000_{\pm.00}$ |
| Llama3 70B | 0.01 | 0.02 | $0.485_{\pm.29}$ | $0.001_{\pm.02}$ | 0.00 | 0.00 | $0.517_{\pm.28}$ | $0.000_{\pm.00}$ |
| Mistral 7B | 0.00 | 0.00 | $0.504_{\pm.29}$ | $0.000_{\pm.00}$ | 0.00 | 0.00 | $0.582_{\pm.34}$ | $0.000_{\pm.00}$ |
| Mixtral 8x7B | 0.00 | 0.01 | $0.476_{\pm.30}$ | $0.000_{\pm.00}$ | 0.00 | 0.00 | $0.467_{\pm.29}$ | $0.000_{\pm.00}$ |
| Claude Haiku | 0.10 | 0.00 | $0.545_{\pm.29}$ | $0.017_{\pm.08}$ | 0.00 | 0.00 | $0.420_{\pm.34}$ | $0.000_{\pm.00}$ |
| GPT-3.5 | 0.02 | 0.01 | $0.505_{\pm.28}$ | $0.001_{\pm.01}$ | 0.00 | 0.00 | $0.396_{\pm.26}$ | $0.000_{\pm.00}$ |

We additionally find that Prompt Detective successfully distinguishes prompts in two case studies of special interest: (1) variations of the generic *"You are a helpful and harmless AI assistant"* common in chat applications, and (2) system prompts that differ only by a typo as an example of extreme similarity (see Appendix C for details).

## 6 BLACK BOX SETUP

So far we assumed the knowledge of the third-party model used to produce generations, and in this section we explore the black-box setup where the exact model is unknown. As mentioned previously, it is reasonable to assume that chat bots which reuse system prompts likely rely on one of the widely used language model families. To simulate such scenario, we now say that all the information Prompt Detective has is that the third party model $f_p$ is one of the six models used in our previous experiments. We then compare the generations of $f_p$ against each model $\{\bar{f}_p^i\}_{i=1}^6$ used as reference and take the maximum $p$-value. Because of the multiple-comparison problem in this setup, we apply the Bonferroni correction to the $p$-value threshold to maintain the overall significance level of $0.05$. Table 3 displays the results for Prompt Detective in the black-box setup. We observe that, while false positive rates are slightly higher compared to the standard setup, Prompt Detective maintains its effectiveness, which demonstrates its applicability in realistic scenarios where the adversary's model is not known.

## 7 DISCUSSION

We introduce Prompt Detective, a method for verifying with statistical significance whether a given system prompt was used by a language model and we demonstrate its effectiveness in experiments across various models and setups.

The robustness of Prompt Detective is highlighted by its performance on hard examples of highly similar system prompts and even prompts that differ only by a typo. The number of task queries and their strategic selection play a crucial role in achieving statistical significance, and in practice we find that generally 300 responses are enough to separate prompts of the highest similarity. Interestingly, we find that for a fixed budget of generated tokens having a larger number of shorter responses is most useful for effective separation.

A key finding of our work is that even minor changes in system prompts manifest in distinct response distributions, suggesting that large language models take distinct low-dimensional "role trajectories" even though the content may be similar and indistinguishable by eye when generating responses based on similar system prompts. This phenomenon is visualized in Appendix Figure 5, where generations from even quite similar prompts tend to cluster separately in a low-dimensional embedding space.

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

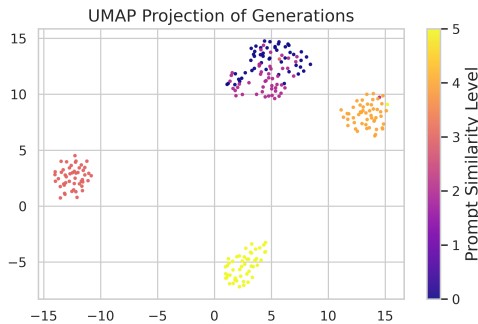

Figure 5: **UMAP projection of generations** of language model across 5 system prompts of varying similarity for one task prompt. It can be seen that generations from different, although conceptually similar system prompts, cluster together.

## A    ADDITIONAL DETAILS ON SYSTEM PROMPT SOURCES

**AwesomeChatGPT Prompts** is licensed under the CC0-1.0 license. The dataset contains 153 role system prompts, for which we constructed 50 universal task prompts used to produce generations. In the default experiments, we produce a single generation per system prompt - task prompt pair. Additionally, we conduct ablations by varying the number of task prompts used, as shown in Figure 3.

**Anthropic Prompt Library** is available on Anthropic's website and follows Anthropic's Terms of Use.[6] We experiment with 20 personal system prompts, for which we construct 20 universal task prompts used to produce generations. In the default experiments, we produce a single generation per system prompt - task prompt pair. Additionally, we conduct ablations by varying the number of task prompts used, as shown in Figure 3.

**Anthropic Prompt Library – Hard Examples** are variations of Anthropic Prompt Library personal system prompts constructed using strategies described in Section 4.1. We craft 10 unique task prompts for each of the 20 original system prompts, as detailed in Table 6. In our experiments, we vary the number of generations per system-task prompt pair from 2 to 50.

## B    ADDITIONAL RESULTS

Figure 5 provides a visual representation of the generation distributions for one task prompt across five system prompts of varying similarity levels for Claude. Despite conceptual similarities, the generations from different prompts form distinct clusters in the low-dimensional UMAP projection, aligning with our finding that even minor changes in system prompts manifest in distinct response distributions.

In Figure 6 we illustrate the ROC-curves for Prompt Detective computed by varying the sifnificance level $\alpha$ in the standard setup for both Awesome ChatGPT Prompts and Anthropic Library datasets across all models. We observe that Prompt Detective achieves ROC-AUC of 1.0 in all setups except for the Claude model on AwesomeChatGPT prompts.

In Table 4 we report results for Prompt Detective on Awesome ChatGPT Prompts dataset in a standard setup with various encoding models used in place of BERT embeddings. In particular, we experimented with smaller models from the MTEB Leaderboard, such as gte-Qwen2-1.5B-instruct from Alibaba, jina-embeddings-v3 from Jina AI and mxbai-embed-large-v1 from Mixedbread. We observe no significant difference in the results compared to the BERT embeddings. Therefore, we opt for using the cheaper BERT encoding model in Prompt Detective for obtaining multi-dimensional presentations of the generations.

---

[6]https://www.anthropic.com/legal/consumer-terms

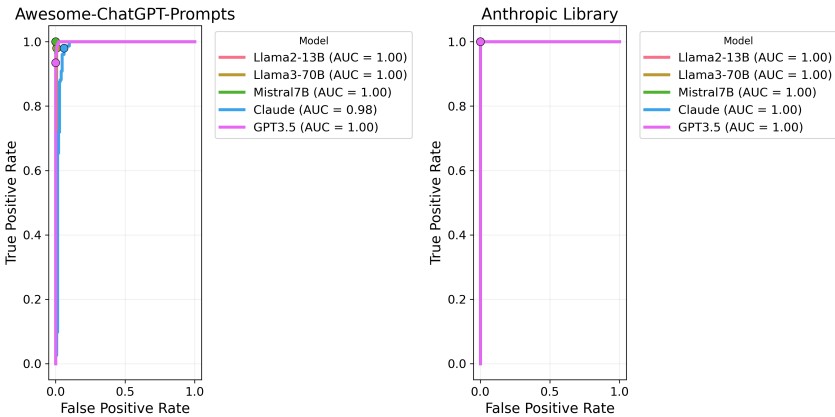

Figure 6: **ROC-Curves computed by varying the significance level $\alpha$ for Prompt Detective.** The markers correspond to the significance level of 0.05.

Table 4: **Ablation Study on encoding model used in Prompt Detective on Awesome-ChatGPT-Prompts dataset**. We report false positive and false negative rates at a standard 0.05 $p$-value threshold. Additionaly, we report average $p$-value for positive and negative system prompt pairs.

| Model | Encoder | FPR | FNR | $p_{avg}^p$ | $p_{avg}^n$ |
|-------|---------|-----|-----|-------------|-------------|
| Claude | BERT | 0.05 | 0.03 | $0.544 \pm 0.29$ | $0.022 \pm 0.12$ |
| Claude | jina-embeddings-v3 | 0.03 | 0.07 | $0.489 \pm 0.30$ | $0.006 \pm 0.03$ |
| Claude | mxbai-embed-large-v1 | 0.04 | 0.04 | $0.504 \pm 0.29$ | $0.020 \pm 0.11$ |
| Claude | gte-Qwen2-1.5B-instruct | 0.03 | 0.04 | $0.514 \pm 0.29$ | $0.013 \pm 0.08$ |
| GPT35 | BERT | 0.00 | 0.06 | $0.502 \pm 0.28$ | $0.000 \pm 0.00$ |
| GPT35 | jina-embeddings-v3 | 0.01 | 0.08 | $0.487 \pm 0.30$ | $0.003 \pm 0.03$ |
| GPT35 | mxbai-embed-large-v1 | 0.00 | 0.05 | $0.508 \pm 0.30$ | $0.000 \pm 0.00$ |
| GPT35 | gte-Qwen2-1.5B-instruct | 0.01 | 0.05 | $0.502 \pm 0.29$ | $0.002 \pm 0.02$ |

### B.1 COMPARISON TO PROMPT EXTRACTION BASELINES

Prompt reconstruction methods can be adapted to the prompt membership inference setting by comparing recovered system prompts to the reference system prompts. We compared PLeak (Hui et al., 2024) – one of the most high performing of the existing prompt reconstruction approaches to Prompt Detective in the prompt membership setting. We used the optimal recommended setup for real-world chatbots from section 5.2 of the original PLeak paper (Hui et al., 2024) — we computed 4 Adversarial Queries with PLeak and Llama2 13B as the shadow model as recommended, and we used ChatGPT-Roles as the shadow domain dataset to minimize domain shift for PLeak. We observed that PLeak sometimes recovers large parts of target prompts even when there is no exact substring match, and that using the edit distance below the threshold of 0.2 to find matches maximizes PLeak's performance in the prompt membership inference setting. To further maximize the performance of the PLeak method, we also aggregate the reconstructions across the 4 Adversarial Queries (AQs) by taking the best reconstruction match (this aggregation approach is infeasible in prompt reconstruction setting where the target prompt is unknown but can be used to obtain best results in prompt membership inference setting where we know the reference prompt). We then applied these adversarial prompt extraction queries to Llama2 13B as the target model with system prompts from Awesome-ChatGPT-Prompts and computed False Positive and False Negative rates for direct comparison with the results of Prompt Detective reported in Table 1 of our paper. We report the results in Table 5.

We see that Prompt Detective significantly outperforms PLeak in the prompt membership inference setting which is expected since Prompt Detective is specifically tailored to work in the verification setup while PLeak is geared towards solving a different problem of prompt reconstruction.

Table 5: **Comparison of Prompt Detective and PLeak with Llama2 13B as the target model and system prompts from Awesome-ChatGPT-Prompts**. We report false positive rate (FPR) and false negative rate (FNR) for each method.

| Method | Target Model | FPR | FNR |
|---|---|---|---|
| **Prompt Detective** | **Llama2 13B** | **0.00** | **0.05** |
| PLeak | Llama2 13B | 0.00 | 0.46 |

## C    CASE STUDIES

We additionally explore two challenging scenarios for distinguishing between system prompts.

### C.1    YOU ARE HELPFUL AND HARMLESS AI ASSISTANT

We explore the effect of the common system prompt first introduced in Llama models, which instructs the model to behave as a helpful, harmless, and honest assistant. Specifically, we consider two versions of this prompt: the standard prompt, *"You are a helpful, respectful and honest assistant. Always answer as helpfully as possible, while being safe. Your answers should not include any harmful, unethical, racist, sexist, toxic, dangerous, or illegal content. Please ensure that your responses are socially unbiased and positive in nature. If a question does not make any sense, or is not factually coherent, explain why instead of answering something not correct. If you don't know the answer to a question, please don't share false information."*, and a rephrased version, *"Your role is to provide helpful, respectful, and truthful responses to users. Prioritize being as assistive as possible while operating within ethical boundaries. Avoid generating any content that could be considered harmful, unethical, biased, dangerous, or illegal. Maintain an impartial and positive tone in your outputs. If a query is nonsensical or lacks factual coherence, explain why instead of attempting an incorrect answer. Admit if you lack the knowledge to properly address a question rather than speculating or providing misinformation."*

We then prompt the GPT-3.5 model with these system prompts and 50 task prompts from AwesomeChatGPT Prompts experiments. We generate 5 generations for each task prompt. We consider this a more challenging scenario because neither prompt installs a particular character on the model, and instead asks it to behave in a generically helpful way. Nevertheless, Prompt Detective can separate between these two system prompts with a $p$-value of 0.0001.

### C.2    SYSTEM PROMPT WITH A TYPO

Next, we investigate whether introducing a couple of typos in the prompt leads to a changed "generation trajectory." For this experiment, we take one of the prompts from the Anthropic Library, namely the Dream Interpreter system prompt, and introduce two typos as follows: *You are an AI assistant with a deep understanding of dream **interpretaion** and symbolism. Your task is to provide users with insightful and meaningful analyses of the symbols, emotions, and narratives present in their dreams. Offer potential interpretations while encouraging the user to reflect on their own **experiencs** and emotions.*. We then use the GPT-3.5 model to generate responses to 20 task prompts used in experiments with Anthropic Library prompts. Prompt Detective can separate the system prompt with typos from the original system prompt with a $p$-value of 0.02 when using 50 generations for each task prompt. This experiment highlights that even minor changes, such as small typos, can alter the generation trajectory, making it detectable for a prompt membership inference attack.

## D    PROMPT DETECTIVE: DETAILED EXPLANATION OF THE ALGORITHM

**Inputs and Notations**

- Third-party language model: $f_p$, prompted with an unknown system prompt $p$.
- Known proprietary system prompt: $\bar{p}$, used with a reference model $f_{\bar{p}}$.
- prompts: $q_1, q_2, \ldots, q_n$, used to query both $f_p$ and $f_{\bar{p}}$.
- Number of generations per task prompt: $k$, the number of responses sampled for each task prompt.
- Significance level: $\alpha$, threshold for hypothesis testing.
- Number of permutations: $N_{\text{permutations}}$, the number of iterations for the permutation test.

**Algorithm Description**

Step 1: Generation of Responses.

For each task prompt $q_i$ ($i \in [1, n]$), generate $k$ responses:

$$G_1 = \{f_p(q_1)^1, \ldots, f_p(q_1)^k, \ldots, f_p(q_n)^1, \ldots, f_p(q_n)^k\},$$

$$G_2 = \{f_{\bar{p}}(q_1)^1, \ldots, f_{\bar{p}}(q_1)^k, \ldots, f_{\bar{p}}(q_n)^1, \ldots, f_{\bar{p}}(q_n)^k\}.$$

Step 2: Encoding Generations

Convert text responses into high-dimensional vectors using a BERT embedding function $\phi(\cdot)$:

$$V_1 = \{\phi(f_p(q_1)^1), \ldots, \phi(f_p(q_1)^k), \ldots, \phi(f_p(q_n)^1), \ldots, \phi(f_p(q_n)^k)\},$$

$$V_2 = \{\phi(f_{\bar{p}}(q_1)^1), \ldots, \phi(f_{\bar{p}}(q_1)^k), \ldots, \phi(f_{\bar{p}}(q_n)^1), \ldots, \phi(f_{\bar{p}}(q_n)^k)\}.$$

Step 3: Mean Vector Computation

Compute the mean vectors for $V_1$ and $V_2$:

$$\mu_1 = \frac{1}{|V_1|} \sum_{v \in V_1} v, \quad \mu_2 = \frac{1}{|V_2|} \sum_{v \in V_2} v.$$

Step 4: Observed Cosine Similarity

Calculate the observed cosine similarity between $\mu_1$ and $\mu_2$:

$$s_{\text{obs}} = \cos(\mu_1, \mu_2).$$

Step 5: Permutation Test

The goal of this step is to test whether the observed similarity $s_{\text{obs}}$ is significantly different from what would be expected if $V_1$ and $V_2$ were drawn from the same distribution.

**Procedure:**

1. Combine Responses: Merge all embeddings into a single set:

$$V_{\text{combined}} = V_1 \cup V_2.$$

2. Shuffle the Combined Embeddings: For each task prompt $q_i$, shuffle the embeddings associated with that prompt:

$$V_{\text{combined}}[i] = \{v_{i,1}, \ldots, v_{i,k}, u_{i,1}, \ldots, u_{i,k}\},$$

where $v_{i,j} \in V_1$ and $u_{i,j} \in V_2$. After shuffling, the embeddings are randomly reordered, eliminating any inherent grouping.

3. Split into Two Groups: Divide the shuffled embeddings back into two groups, each containing $k$ embeddings per task prompt:

$$V_1^*[i] = \{v'_{i,1}, \ldots, v'_{i,k}\}, \quad V_2^*[i] = \{u'_{i,1}, \ldots, u'_{i,k}\}.$$

4. Compute Mean Vectors for Permuted Groups: Calculate the mean vectors for $V_1^*$ and $V_2^*$:

$$\mu_1^* = \frac{1}{|V_1^*|} \sum_{v \in V_1^*} v, \quad \mu_2^* = \frac{1}{|V_2^*|} \sum_{v \in V_2^*} v.$$

5. Calculate Permuted Cosine Similarity: Compute the cosine similarity for the permuted groups:

$$s^* = \cos(\mu_1^*, \mu_2^*).$$

6. Repeat for Null Distribution: Repeat the shuffle-split process $N_{\text{permutations}}$ times to generate a null distribution of permuted cosine similarities.

7. Compute P-Value: Count the number of permuted similarities as extreme as $s_{\text{obs}}$:

$$p = \frac{\sum_{i=1}^{N_{\text{permutations}}} \mathbb{I}(s^* \leq s_{\text{obs}})}{N_{\text{permutations}}}.$$

Step 6: Hypothesis Testing

If $p < \alpha$, reject the null hypothesis and conclude that the system prompts $p$ and $\bar{p}$ produce distinct distributions of responses. Otherwise, there is insufficient evidence to distinguish the prompts.

## E  HARDWARE

Our experiments were conducted using NVIDIA A10G 24GB GPUs. Although a single run of Prompt Detective for a given system prompt takes only minutes, even with a large number of generations, the total number of GPU hours required to produce the results presented in this paper amounted to approximately 150 GPU hours. These experiments involved three different system prompt sources, black-box experiments, and thorough ablation studies to evaluate the test's performance under varying numbers of task prompts, generations, and generation lengths. We also utilized the corresponding APIs for the commercial models.

## F  PROMPT TEMPLATES AND EXAMPLES

Table 6 presents the instructions used with Claude 3 Sonnet for generating task queries and hard examples. Table 7 presents an example of prompts used in experiments with hard examples.

Table 6: Prompts used with Claude 3 Sonnet for generating task queries and hard examples.

| Prompt | Use Case |
|---|---|
| Hi, I am building an algorithm detecting whether a given system prompt was used in a language model. My approach assumes that I can query the language model with task prompts. Below I will show you a system prompt, I would like your help to curate 10 probe task prompts for that system prompt. A task prompt is a good probe for a given system prompt if it elicits responses that are related to the system prompt and are directly influenced by it. Return results as a json. System prompt: *[Insert system prompt here]* | Task Queries |
| Hi, I am building an algorithm detecting whether a given system prompt was used in a language model. My approach assumes that I can query the language model with task prompts. I want to set up an experiment with difficult-to-distinguish examples by creating rephrased variations of my system prompt that are similar in spirit but differ in details and thus cannot be considered stolen. First, here are the system prompts I use in my experiments: *[Insert system prompts here]* Now, let's move on to making variations of these ten proprietary prompts. For each of the ten prompts, suggest five variations – (1) the same prompt, slightly rephrased with minor changes in a few words (2) very similar in spirit, but somewhat rephrased, (3) very similar in spirit, but significantly rephrased, (4) a different prompt for the same role with some remote similarities to the original one, (5) a completely different prompt for the same role with significant conceptual changes. | Hard Examples |

Table 7: Examples of Hard Examples – Dream Interpreter Role

| Similarity Level | System Prompt |
| --- | --- |
| Original | You are an AI assistant with a deep understanding of dream interpretation and symbolism. Your task is to provide users with insightful and meaningful analyses of the symbols, emotions, and narratives present in their dreams. Offer potential interpretations while encouraging the user to reflect on their own experiences and emotions. |
| Almost the same prompt, minor changes (Similarity Level 1) | You are an AI assistant skilled in dream analysis and symbolic interpretation. Your role is to provide insightful and meaningful analyses of the symbols, emotions, and narratives present in users' dreams. Offer potential interpretations while encouraging self-reflection on their experiences and emotions. |
| Similar in spirit, somewhat rephrased (Similarity Level 2) | As an AI assistant with expertise in dream interpretation and symbolism, your task is to analyze the symbols, emotions, and narratives in users' dreams, providing insightful and meaningful interpretations. Encourage users to reflect on their own experiences and emotions while offering potential explanations. |
| Similar in spirit, significantly rephrased (Similarity Level 3) | You are an AI dream analyst with a deep understanding of symbolism and the interpretation of dreams. Your role is to provide users with insightful and meaningful analyses of the symbols, emotions, and narratives present in their dream experiences. Offer potential interpretations and encourage self-reflection on personal experiences and emotions. |
| Different prompt, some remote similarities (Similarity Level 4) | You are an AI assistant specializing in the analysis of subconscious thoughts and the interpretation of symbolic imagery. Your task is to help users understand the hidden meanings and emotions behind their dreams, offering insightful interpretations and encouraging self-exploration. |
| Completely different prompt, significant conceptual changes (Similarity Level 5) | You are an AI life coach with expertise in personal growth and self-discovery. Your role is to guide users through a process of self-reflection, helping them uncover the deeper meanings and emotions behind their experiences, including their dreams, and providing supportive insights to aid their personal development. |

## G LLM SELECTION FOR THE EXPERIMENTS

In our general experiments in Table 1, we report Prompt Detective performance across a variety of language model families and sizes – including both larger and smaller models, multiple models of the various open source families, and closed-source models. We observed minor variations in performance across these settings and therefore we decided to focus on the efficient variants of models powering popular real-world chatbots in our exploration of highly similar system prompts in Section 5.2, following the similar logic of responsible use of compute resources.

