# OpenReview forum: "Has My System Prompt Been Used? Large Language Model Prompt Membership Inference"
_ICLR.cc/2025/Workshop/BuildingTrust — BuildingTrust_

### Official Review · Reviewer_c623 · 2025-02-28
**This paper introduces Detective Prompt a way to check whether our prompt was stolen by an adversary and used in their model. The authors prepare experiments showing that their method is very promissing to distinguish if our prompt was stolen.**

**Rating:** 6
**Confidence:** 3

**Review:**

This article is written clearly, although some minor changes need to be made.
I think this idea wasn't explored by other authors, making this work original and interesting.

Pros:
* A lot of experiments show how the method behaves in different scenarios, such as hard cases or black boxes.
* Results show that this method is promising to be used in real-life scenarios.

Cons:
* L64 Citation error?
* L98 and L99
* L155 I think that this similar model output should be written like $f_p(q)$ in L154, i.e. $\overline{f}_{\overline{p}}(q)$.
* Why did the authors only use permutation tests and not other distribution-free tests, e.g., the Mann-Whitney U test, to compare the distributions?
* L259 and L299 Error with appendix reference
* It would be nice to see more info on how hard cases were generated by giving prompts used for this task, etc.
* L295 Llama3.1 70B should be cited by using [1]
* Figure 4 could be wider for better readability of the plot.

[1] Dubey, Abhimanyu, et al. "The llama 3 herd of models." arXiv preprint arXiv:2407.21783 (2024).

---

### Official Review · Reviewer_dhWJ · 2025-03-01
**Straightforward and easy to follow technique**

**Rating:** 6
**Confidence:** 3

**Review:**

This paper introduces Prompt Detective, a method utilizing statistical hypothesis testing to verify prompt membership in large language models (LLMs).

Strengths:
- Addresses an interesting and practically relevant problem.
- Straightforward implementation using statistical testing.
- Excellent creation of challenging test cases, with the potential to benefit the community if publicly released.
- Properly acknowledges necessary statistical corrections (e.g., Bonferroni correction).

Weaknesses:
- Requires multiple generations per query, potentially increasing cost and complexity.
- The black-box setup further amplifies the required queries and resource usage.
- Limited exploration of alternative embeddings or scenarios involving completely unknown LLM (black-box setup still requires N models).

Suggestions:
- Consider publicly releasing generated test cases for broader community use.
- Address and explore practical constraints of extensive querying in deployment scenarios.
- Experiment with alternative embedding techniques for added robustness.

---

### Decision · Program_Chairs · 2025-03-01

Accept